# Development and external validity of a nurse-led intervention program to improve palliative care and quality of dying and death in intensive care unit

Kazuaki Naya[1], Hideaki Sakuramoto[2]*

**1** Department of Adult Health Nursing, Tokyo Healthcare University, Wakayama Faculty of Nursing, Wakayama City, Wakayama, Japan, **2** Division of Faculty Development, Department of Nursing, Kindai University, Sayama, Osaka, Japan

* gongehead@yahoo.co.jp

## Abstract

### Introduction

In Japan, end-of-life care in intensive care units (ICU) remains inconsistent and often suboptimal, contributing to variability in the quality of dying and death. This study aimed to develop a culturally appropriate nurse-led intervention program to improve the quality of dying and death in Japanese ICU.

### Methods

Using a structured framework for intervention development, we conducted a systematic and scoping review to identify modifiable factors and effective strategies for end-of-life care in ICUs. Eight evidence-based strategies were synthesized into four core components: symptom management, multidisciplinary bedside conferences, family conferences, and end-of-life care. A preliminary intervention program was developed and evaluated through expert interviews and web-based surveys to assess its clarity, feasibility, and alignment with the Japanese healthcare system.

### Results

Based on expert feedback, revisions were made to enhance the clarity, feasibility, and alignment with the Japanese healthcare system. Key improvements included initiating symptom management early upon ICU admission, allowing flexible scheduling of conferences, and incorporating structured tools and frameworks commonly used in Japan. Additionally, a ladder-based system was introduced to tailor the intervention intensity according to the patients' clinical conditions.

**Data availability statement:** All relevant data are within the manuscript and its Supporting Information files.

**Funding:** This study was supported by a Grant-in-Aid for Scientific Research (C) from the Japan Society for the Promotion of Science (Grant number 25K13875). The funders had no role in study design, data collection and analysis, decision to publish, or preparation of the manuscript.

**Competing interests:** The authors have declared that no competing interests exist.

## Conclusion

The final nurse-led intervention program was designed to facilitate more consistent delivery of end-of-life care in the ICU regardless of individual providers' knowledge or attitudes. This study demonstrates a rigorous and culturally adapted intervention development process that may serve as a model for improving end-of-life care in ICU in Japan and other healthcare systems.

## Introduction

The intensive care unit (ICU) is primarily designed to treat patients with critical illnesses who have the potential for recovery; however, owing to illness severity, ICU mortality remains high. In settings where life-saving interventions are prioritized, it is often difficult to create an appropriate end-of-life (EOL) environment, and adequate EOL care is not consistently provided [1]. A comparison of the Quality of Dying and Death (QODD) assessed by bereaved family proxies between ICU and hospice settings revealed that patients in the ICU have significantly lower QODD scores than those in hospice settings [2]. Notably, scores for pain control (ICU: $2.1 \pm 2.1$; hospice: $5.3 \pm 3.1$; $P = 0.001$) and breathing comfort (ICU: $2.1 \pm 2.7$; hospice: $4.1 \pm 3.0$; $P = 0.032$) were markedly lower in the ICU group than in the hospice group. The key factors associated with QODD include the presence of advanced directives, effective communication with families, and support for decision making [3].

In the ICU setting, patient prognosis at admission is often uncertain, and may change over time. Many patients who die in the ICU initially receive aggressive, life-sustaining treatments with the expectation of recovery. Therefore, discussions regarding patient values, preferences, and advance directives should be initiated early after ICU admission and revisited as clinical conditions evolve. Inadequate symptom control is a major contributor to poor QODD [4], and ICU patients frequently experience significant unrelieved physical and psychological distress [5]. Accordingly, palliative care should be provided to all ICU patients regardless of their prognosis. Although ICU palliative care initially focused on EOL care, it is now understood to encompass symptom management, shared decision-making, and support for patients and their families throughout the course of the critical illness [6].

Although various interventions have attempted to improve QODD, few have systematically integrated multiple strategies to address its complex determinants [7]. In Japan, research focusing on the EOL care for ICU patients and their families is scarce. One study reported that patients with terminal cancer in Japanese ICUs received lower-quality EOL care than those in palliative care units, citing insufficient knowledge and skills among physicians and nurses [8]. Additionally, a multicenter cross-sectional study found that the use of a high-flow nasal cannula may improve symptom relief and QODD at the EOL [9]. Despite these insights, culturally and systematically appropriate interventions for ICU EOL care remain underdeveloped in Japan.

This study aimed to develop an intervention program to improve QODD in Japanese ICUs, guided by the Six Steps in Quality Intervention Development (6SQuID) framework [10] and adapted to Japan's healthcare system and cultural context.

## Methods

The intervention program was systematically developed based on the first four steps of the 6SQuID framework [10], as illustrated in Fig 1. Initially, we conducted a systematic review and meta-analysis (SR/MA) [3] and a scoping review (ScR) [7]. The findings from these reviews were synthesized and applied in Steps 1–3. Step 4 involved drafting a preliminary version of the intervention program and assessing its internal and external validity. Based on the results, the program was revised and finalized.

The study protocol was approved by the Ethics Committee of the Japanese Red Cross Kyushu International College of Nursing (Approval No. 24−014, Approval Date: July 3, 2024). The participants were recruited between July 3 and July 11, 2024. All participants gave consent to participate before starting the focus group interviews (FGI) and Web-based surveys. Reviewing the explanatory document and submitting the consent form were regarded as provision of consent. They were also informed that their participation was voluntary and that they could withdraw at any time without penalty.

### Literature review of QODD in the ICU

**STEP 1: Define and understand the problem and its causes.** A review of the existing literature revealed a clear need for both clinical and academic interventions to improve QODD in ICU settings. The underlying causes of suboptimal EOL care were analyzed to identify the key contributing factors.

**STEP 2: Clarify which causal or contextual factors are malleable and have the greatest scope for change.** We identified the modifiable causes and contextual factors with the highest potential for impact. In cases where causal

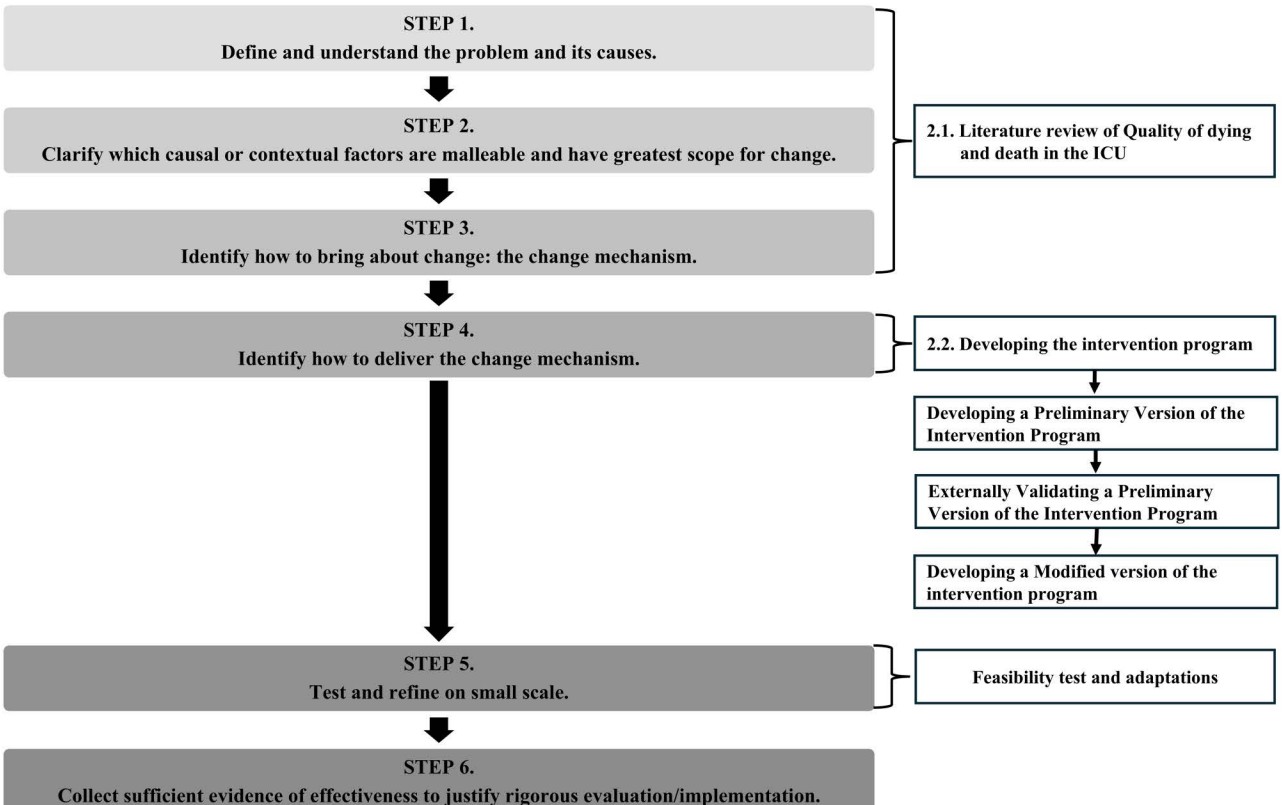

**Fig 1. Process for developing a preliminary version of the intervention program.**

pathways were complex and interrelated, we pinpointed critical intervention points either to disrupt causal chains or apply multilevel strategies. We also assessed the changes likely to produce the greatest improvements in QODD.

**STEP 3: Identify how to bring about change (the change mechanism).** For each modifiable factor, feasible strategies for change were examined. The mechanisms of action were clearly defined to illustrate how each intervention component contributed to the improved QODD outcomes.

### Developing the intervention program

**Developing a preliminary version of the intervention program. STEP 4: Identify how to deliver the change mechanism.** A preliminary intervention program was constructed by incorporating the change mechanisms identified in Step 3. To assess validity, both internal and external evaluations were conducted through FGIs with experts and a web-based expert survey.

**Externally validating a preliminary version of the intervention program.**

(1) **FGI with experts**

An FGI was conducted to evaluate the internal and external validity of the preliminary program. Snowball sampling was used to recruit the experts. Individuals were eligible to participate if they met the inclusion criteria (a), (b), or (c) and did not meet any of the exclusion criteria. The eligibility criteria were as follows: (a) physicians or nurses with at least five years of experience working in ICUs; (b) nurses certified in intensive care, certified nurse specialists in critical care nursing, educators, and researchers specializing in intensive care; and (c) physicians involved in palliative care or EOL care in the ICU. The exclusion criteria were as follows: (d) participants with no experience in palliative or EOL care in the ICU, (e) participants with a conflict of interest related to this study, (f) participants who were unable to provide informed consent, and (g) participants who had experienced the loss of a family member within the past six months. Since early bereavement reactions and psychological symptoms tend to be most pronounced within the first six months following loss [11], these individuals were excluded to minimize psychological distress and avoid triggering recollection of recent bereavement.

Interviews were conducted remotely via videoconferencing. Audio recordings were transcribed verbatim and reviewed multiple times. Common themes were identified and synthesized. Revisions of the preliminary program were based on expert feedback. Qualitative saturation was achieved when no new themes or suggestions relevant to the refinement of the intervention program emerged during the later stages of the FGI. To ensure trustworthiness of the content analysis, the data were reviewed repeatedly by two researchers and supervised by a qualitative research expert. The raw data were rechecked to ensure accuracy. Additionally, the analyzed data were verified with the participants to ensure that their statements were accurately reflected and that no additional opinions were omitted. The revised version was defined as the "modified intervention program." The interview guidelines are presented in Table 1.

(2) **Web-based survey with experts**

Content validity of the modified program was assessed using a web-based expert survey. The survey items are presented in Supporting Information (see S1 Text). The process was as follows: (a) Experts were provided with the FGI data together with the modified program and then asked to evaluate the alignment between the revisions and FGI findings. (b) Quantitative evaluation was conducted using Lynn's method [12] and the content validity index (CVI) was calculated. Qualitative comments were also collected. (c) Items were rated on a four-point scale (4 = valid, 3 = fairly valid, 2 = somewhat lacking validity, and 1 = not valid). The item-level CVI (I-CVI) was calculated as the proportion of items with a score of three or four, with ≥0.78 set as the threshold for acceptable content validity. (d) Items with I-CVI < 0.78 were revised using qualitative feedback. This evaluation–modification cycle was repeated until all items reached I-CVI ≥ 0.78.

**Table 1. Interview guide.**

| Question |
| --- |
| (1) The intervention program includes care for end-of-life patients and for others. What are your thoughts in this regard? |
| (2) If you were to provide care according to the intervention program, are there any aspects you find challenging to implement or consider that they should be revised? |
| (3) Suppose you were responsible for developing this program and had the authority to improve it. What would be the first thing you would change? Please focus on modifications to the existing content rather than additions. |
| (4) Among the topics discussed today, what did you find most important? |
| (5) Following a brief verbal summary of the interview, does the summary accurately reflect what was discussed? |
| (6) After restating the purpose of the study, do you have any additional comments or is there anything you consider was overlooked during the interview? |

## Sample size for FGI

Based on a previous study [13], we targeted a focus group of six to eight participants.

## Results

### Literature review of QODD in the ICU

**STEP 1: Define and understand the problem and its causes.** The SR/MA results identified multiple factors associated with improved QODD in the ICU. These included the absence of cardiopulmonary resuscitation at EOL, death without the use of full life-sustaining measures, provision of a therapeutic environment, effective symptom management, and preservation of patient dignity (e.g., optimal control of physical symptoms, management of events, and maintenance of self-respect). These findings suggest that interventions aimed at promoting dignity and comfort among ICU patients may improve the quality of EOL care in this setting [3].

**STEP 2: Clarify which causal or contextual factors are malleable and have greatest scope for change.** The ScR identified 10 intervention strategies to improve QODD [7]. Among these, "family participation in bedside rounds" and "feedback on EOL care for healthcare workers" were excluded because of insufficient supporting evidence, thus resulting in the selection of eight strategies. A substantial gap in evidence has been noted regarding symptom management, underscoring the need to develop intervention programs that specifically target this aspect.

**STEP 3: Identify how to bring about change (the change mechanism).** To improve QODD in the ICU, we identified eight evidence-based intervention strategies, grouped into four core components. For each strategy, we articulated the underlying mechanisms by which these changes were expected to occur.

(1) **Evidence-based symptom management**

Improving the control of distressing symptoms (e.g., pain and agitation) reduces physical suffering, thereby directly enhancing patient comfort and dignity, which are the core domains of QODD.

(2) **Multidisciplinary bedside conference**

Regular interdisciplinary communication fosters shared decision-making and early palliative care. This improves goal-concordant care and symptom management, leading to improved QODD.

(3) **Family conference**

Early and structured family meetings support informed surrogate decision making, thus reducing unnecessary invasive treatments, and increasing family satisfaction. These are correlated with an improved QODD.

## (4) EOL care

Emotional support and honoring patient/family wishes during the dying process reduce treatment intensity, align care with values, and preserve dignity, thus leading to improved QODD.

## Developing the intervention program

**Developing a preliminary version of the intervention program. STEP 4: Identify how to deliver the change mechanism.** Based on these eight intervention strategies, we developed a preliminary nurse-led intervention program to improve palliative care and QODD in ICU. Rather than following a ladder-based approach, the intervention was designed with a time- and condition-dependent flow of care. Patients were screened based on their clinical status over time and subsequent steps in the intervention proceeded according to this screening, creating a structured flow throughout the ICU stay. A flow diagram was developed to visually guide the practice.

Given the complexity of transitioning from curative to EOL care in modern ICUs, it was necessary to identify appropriate patients at an early stage, rather than focusing only on those who are near death. Therefore, primary screening criteria were established to identify eligible patients from among all ICU admissions. Once identified, evidence-based symptom management was initiated on ICU admission. For these patients, multidisciplinary bedside conferences were held within 24 h to clarify the goals of care and discuss treatment strategies. This step enabled the early planning of family conferences and timely initiation of EOL care. Nurses closely involved in daily patient care serve as key agents of the program. Their roles in symptom assessment, patient identification, and coordination of multidisciplinary meetings were explicitly defined and formally integrated into the intervention procedures. The intervention program was compiled into a booklet including an intervention flow diagram and designed for distribution to the departments participating in the intervention.

## Externally validating a preliminary version of the intervention program

Six experts conducted the FGI and web surveys. Based on the FGI results, the preliminary program was modified, and the validity of these modifications was assessed through a web survey. The experts included two physicians and four nurses, all of whom were clinical nurses in the ICU as well as educators and researchers in palliative care and symptom management (Table 2).

## Validation by FGI for experts

The expert feedback was categorized into the following areas: scope and selection of target populations for the intervention program (Table 3), strategies to enhance the feasibility of the intervention, incorporation of evidence from Japan, and refinement of terminology and wording (Table 4).

**Table 2. Characteristics of the participants.**

| Variable | n (%) |
|---|---|
| Male | 4 (66.7) |
| Physicians involved in palliative care or EOL care in the ICU | 2 (33.3) |
| Educator and researcher specializing in palliative care in the ICU | 1 (16.7) |
| Educators and researchers specializing in symptom management in the ICU | 1 (16.7) |
| Critical certified nurse specialist | 2 (33.3) |

EOL, end-of-life; ICU, intensive care unit

**Table 3. Expert Opinions on the Target Population of the Intervention Program.**

| Topic | Expert opinions | Main revisions |
|---|---|---|
| Care Recipients Under the Intervention Program | | |
| Scope of the Target Population | Including patients who are not in the terminal stage in a program designed to improve the quality of dying and death was considered inappropriate. | The program content was restructured into a tiered format, with symptom management integrated at each level. As determining when ICU patients transition to the terminal stage is challenging, non-terminal patients remain included. |
| | Symptom management should be provided to all patients, not only those in the terminal stage; therefore, non-terminal patients should be included. | |
| Selection of the Target Population | The initial screening criteria are appropriate because they enable the identification of severely ill patients. | The intervention program flowchart was revised to a tiered format, and a process was added at the initial screening stage for transitioning to end-of-life care. |
| | A process may be needed during the initial screening stage to determine whether end-of-life care is appropriate. | |
| | The item "4. Patients who are not native Japanese speakers" may be unnecessary in the screening process. | (3) The item regarding non-native Japanese speakers was removed from the screening criteria and incorporated into "Basic Information Related to the Quality of End-of-Life Care." |

ICU, intensive care unit

## (a) Scope and selection of target populations for the intervention program

Regarding the selection of target populations for the intervention program, some experts expressed concern about the inclusion of patients with non-terminal illnesses and noted a potential inconsistency with the program's stated aim of improving the QODD. They highlighted that, "Including patients who are not in the terminal stage in a program designed to improve the quality of dying and death was considered inappropriate."

Conversely, there was a consensus on the importance of including patients outside the terminal phase, particularly because the core symptom management component of the intervention program should be provided to all patients in the ICU, and not only to those who are terminally ill. In essence, "Symptom management should be provided to all patients, not only those in the terminal stage; therefore, non-terminal patients should be included."

Regarding patient selection, it was proposed that a primary screening item should be used at the time of ICU admission to identify critically ill patients who may benefit from an intervention program. However, some experts noted the absence of a clear process for determining the transition to EOL care during this initial screening: "The initial screening criteria are appropriate because they enable the identification of severely ill patients. A process may be needed during the initial screening stage to determine whether end-of-life care is appropriate."

The primary screening items were revised in response to these comments, and the intervention process, which originally followed a time- and condition-based flow, was reorganized into a stepwise ladder-like format (Fig 2). This allowed for the initiation of symptom management from the time of ICU admission without altering the overall target population specified in the intervention program.

Following this reorganization, explicit operational criteria were defined for transitions between ladder levels (Fig. 2). Ladder I was initiated for patients who required, or were highly likely to require, mechanical ventilation for more than 48 hours, or who were admitted emergently from a general ward. The transition from Ladder I to Ladder II was considered when the patient's condition deteriorated or the response to treatment was judged to be poor. Nurses conducted this assessment during daily symptom management conferences. The transition from Ladder II to Ladder III was determined

**Table 4. Expert opinions and revisions on the feasibility of the intervention program.**

| Program components | | |
| --- | --- | --- |
| **Topic** | **Expert opinions** | **Main revisions** |
| Provision of Evidence-based Symptom Management | | |
| Methods (Timing, Participants, Procedures, etc.) | Prompt initiation of symptom management upon ICU admission is essential. | "Evidence-based Symptom Management" was revised to begin based on the results of the initial screening conducted upon admission. |
| | Is the frequency of the "Symptom Management and Palliative Care Conference" appropriate? It should be conducted daily. | The conference frequency was revised from once every 3 days to daily, and a note was added stating it can be integrated into existing rounds and conferences. |
| | Given current drug use practices in Japanese ICUs, "morphine" should be replaced with "opioids" for pain management, as fentanyl is often preferred. | References to "morphine" and "fentanyl" were consolidated under the term "opioids." |
| Use of Japanese Evidence | Some listed assessment tools (e.g., the Japanese version of the RDOS) are not widely used in Japan; should alternatives be allowed? | The listed assessment tools are examples; other tools may be used.<br>The ICDSC was added alongside the CAM-ICU as a delirium assessment tool. |
| | The contents of Japan's clinical practice guidelines for dry mouth can serve as a reference. | ROAG and OAG were included based on Japanese clinical guidelines. |
| Revisions to Wording | As this is a nurse-led program, it would be better to revise the sentence subjects to refer to nurses. | Sentence subjects were revised to refer to nurses.(For example: "Suggest … to the attending physician," etc.) |
| Multidisciplinary Bedside Conferences | | |
| Methods (Timing, Participants, Procedures, etc.) | Holding the initial conference within 24 h of ICU admission is difficult for attending physicians. | The intervention program flowchart was revised into a tiered format (Tiers I–III), limiting the target population for interprofessional bedside conferences to Tier II and above.<br>The timing of the initial conference was extended from "within 24 h of ICU admission" to "within 3 days of starting Tier II" to allow for greater flexibility.<br>An addition was made to clarify that existing rounds and conferences may serve as substitutes for this conference. |
| | It is challenging for attending physicians to participate in the conferences while in open ICUs. | |
| | ICU physicians cannot independently set treatment goals without the attending physician's presence. | |
| | In facilities already conducting conferences, how would this conference be differentiated? | |
| | The content should be simplified using a framework such as Jonsen's four-box method. | The content of the conference was structured according to Jonsen's four-box method.<br>To support regular implementation, the content was simplified through the use of a structured framework. |
| | Given the current shortage of interprofessional conferences, the content should be structured to support regular implementation. | |
| Family Conference | | |
| Methods (Timing, Participants, Procedures, etc.) | Conducting family conferences at the designated times can be challenging. | A revision was made to assess the need for family meetings during the interprofessional bedside conferences and to conduct them as early as possible. |
| | Who should participate in the family meetings? | Participant information was added. |
| | A pre-meeting should be held before the family meeting to ensure alignment among healthcare providers. | An addition was made stating that the interprofessional bedside conference, which determines the need for a family meeting, also functions as a pre-meeting. |
| | Healthcare providers may require training in conducting family conferences. | Communication guides were added to support staff training. |
| Revisions to Wording | The phrase "the patient's values regarding life and death" may be too specific and difficult for the families to address. It may be preferable to use the broader term "the patient's values." | The phrase "the patient's values regarding life and death" was revised to "the patient's values." Specific topics to be discussed during the meeting were outlined. |
| | It may be more appropriate to refer to it as a family meeting rather than a family conference. | The terminology was revised from "family conference" to "family meeting." |
| End-of-Life Care | | |

*(Continued)*

**Table 4.** (Continued)

| Program components | | | |
|---|---|---|---|
| Methods (Timing, Participants, Procedures, etc.) | Clear specification is necessary regarding who should provide the care and when. | The purpose, timing, and providers of end-of-life care were added. |
| | The example information provided needs to be revised to reflect ICU patients. | The example information regarding the condition of patients near the end-of-life was revised to better reflect ICU patients. |
| | It may be beneficial to include information regarding the removal of unnecessary lines, the easing of visitation restrictions, and the potential transfer to a general ward. | Information was added regarding the removal of unnecessary lines, the easing of visitation restrictions, and the potential transfer to a general ward. |
| Use of Japanese Evidence | It may be advisable to include information on grief care and psychological support for families, referencing the "Practice Guide for End-of-Life Nursing in Emergency and Critical Care." | Content addressing care before and after bereavement was added, referencing the "Practice Guide for End-of-Life Nursing in Emergency and Critical Care." |

ICU, intensive care unit; RDOS, Respiratory Distress Observation Scale; ICDSC, Intensive Care Delirium Screening Checklist; CAM-ICU, Confusion Assessment Method for the Intensive Care Unit; ROAG, Revised Oral Assessment Guide; OAG, Oral Assessment Guide.

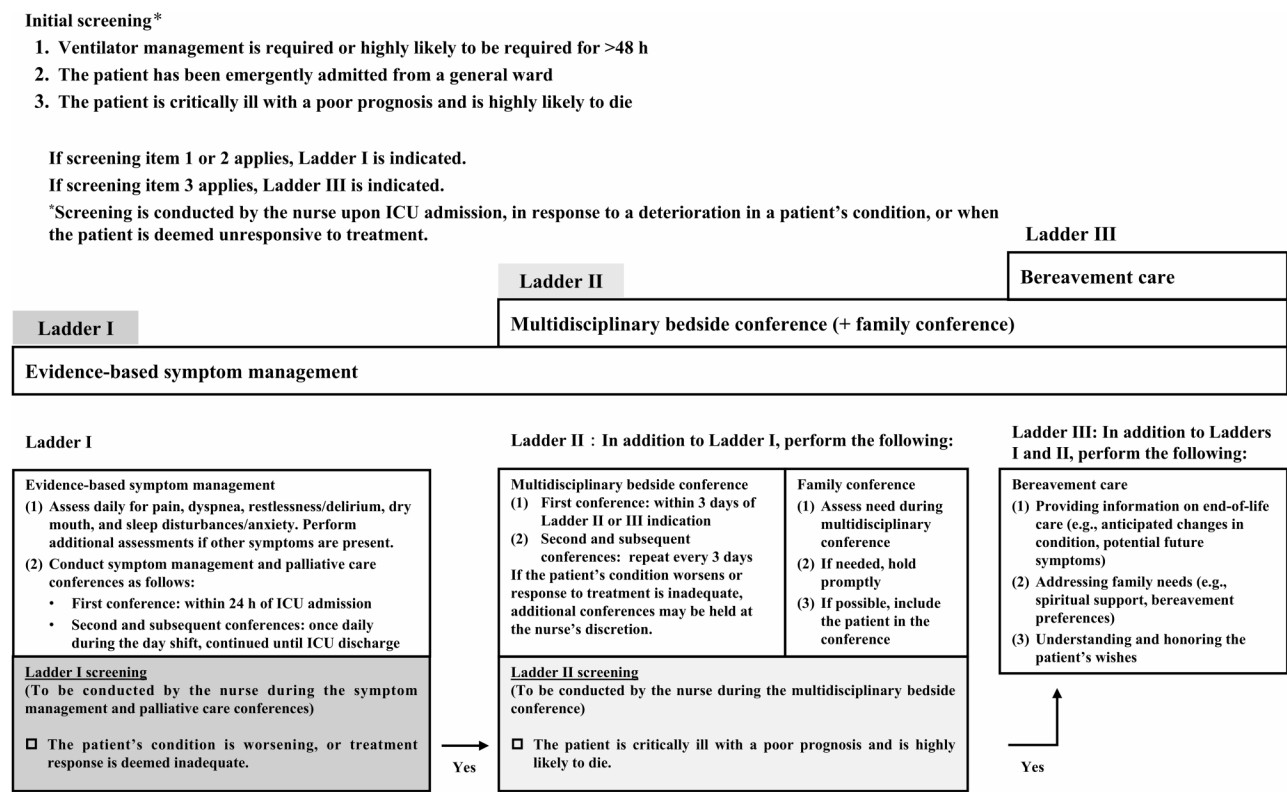

**Fig 2. Quality improvement ladder for palliative care and quality of dying and death in the ICU.**

when life-saving treatment was judged difficult and the likelihood of death was extremely high. This decision was made by the nurses during multidisciplinary bedside conferences. In cases of rapid clinical deterioration or an extremely poor response to treatment, a direct transition from Ladder I to Ladder III was permitted. All screening and transition decisions were nurse-led and reassessed based on changes in the patient's clinical condition.

(b)**Strategies to enhance the feasibility of the intervention and incorporation of evidence in Japan**

Feedback was provided on "evidence-based symptom management," "multidisciplinary bedside conferences," "family conferences," and "EOL care" in relation to strategies for enhancing the feasibility of the intervention, the incorporation of evidence from Japan, and the refinement of terminology and wording.

### Evidence-based symptom management

Regarding symptom management, the importance of initiating daily symptom management early in the ICU stay was emphasized: "Prompt initiation of symptom management upon ICU admission is essential. Is the frequency of the "Symptom Management and Palliative Care Conference" appropriate? It should be conducted daily." In response, the post-screening flowchart was revised to ensure that symptom management was initiated early during ICU admission.

To enable daily symptom management conferences, they were permitted to be incorporated into the existing rounds and meetings. Furthermore, the tools for objective symptom assessment were revised to increase versatility by including instruments that are more commonly used in Japan and avoiding restrictions on the choice of tools.

"Some listed assessment tools (e.g., the Japanese version of the Respiratory Distress Observation Scale) are not widely used in Japan; should alternatives be allowed? The contents of Japan's clinical practice guidelines for dry mouth can serve as a reference."

### Multidisciplinary bedside conference

Owing to variations in ICU systems across facilities (e.g., open ICUs, closed ICUs), some experts noted the challenges related to physician participation in conferences and the difficulty of holding such meetings in a timely manner: "Holding the initial conference within 24 h of ICU admission is difficult for attending physicians. It is challenging for attending physicians to participate in the conferences while in open ICUs. ICU physicians cannot independently set treatment goals without the attending physician's presence."

In response to these concerns, the target population for multidisciplinary bedside conferences was revised from "all patients" to "patients at Ladder II or above." Additionally, the timing of the first conference was modified from "within 24 hours of ICU admission" to "within 3 days of the start of Ladder II," thus allowing for greater flexibility and improving feasibility.

To standardize the content of multidisciplinary conferences and enhance their feasibility, Jonsen's four-topic approach, which is commonly used in Japan, was adopted as the framework. This modification reflected expert feedback, including the following: "The content should be simplified using a framework such as Jonsen's four-box method. Given the current shortage of interprofessional conferences, the content should be structured to support regular implementation."

### Family conference

Challenges regarding the feasibility of a fixed schedule were identified: "Conducting family conferences at the designated times can be challenging." Therefore, the need for a family conference was assessed during the multidisciplinary bedside conference, and the timing was adjusted to "as soon as possible, after the multidisciplinary bedside conference determines that a family conference is necessary," to allow for greater flexibility in implementation.

In addition, the need for education for health care providers regarding communication skills in preparation for holding conferences with families was noted: "Healthcare providers may require training in conducting family conferences." Therefore, a guide for communicating with patients and their families was added as a supplementary document.

## EOL care

In response to expert feedback, the purpose, timing, and responsible providers of EOL care were specified to ensure that interventions are conducted at the appropriate time (i.e., "who will do it and when?").

Additionally, in response to feedback regarding the need to adjust the treatment environment, the EOL care section was revised to include the relaxation of visitation restrictions and consideration of transfer to a general ward. "It may be beneficial to include information regarding the removal of unnecessary lines, the easing of visitation restrictions, and the potential transfer to a general ward."

For family grief care and emotional support, content was added based on the *Practice Guide for End-of-Life Nursing in Emergency and Intensive Care*, which was developed by the Japanese Society of Critical Care Nursing in 2019. "It may be advisable to include information on grief care and psychological support for families, referencing the *Practice Guide for End-of-Life Nursing in Emergency and Critical Care.*"

## Validation via a web-based survey of experts

Following the FGI, the preliminary program was revised and validated using a web-based survey involving the same six experts. The survey was designed to obtain both quantitative assessments of the revised items and qualitative feedback on the proposed modifications. All modified items achieved an I-CVI of 0.78 or higher, thereby confirming their content validity (Table 5). Detailed item-level CVI ratings for each expert are provided in Supporting Information (see S2 Text).

Based on the FGI results, the intervention ladder was categorized into Stages I–III. A process was initially established to transition from Ladder I or II to "Ladder III: End-of-Life Care" when the criterion "The patient or surrogate decision-maker has requested the withholding of treatment" was met. However, in response to qualitative feedback from the experts, this additional criterion was removed. Experts noted that "withholding" includes both the withholding and withdrawal of treatment, and that such decisions do not necessarily reflect an EOL situation.Consequently, the screening criterion for transitioning to Ladder III was simplified to a single item: "Difficult to save and very likely to die." Further expert feedback highlighted that the term "surrogate decision-maker" is not legally defined in Japan, and that a "key person" does not necessarily equate to a surrogate decision-maker. In this context, it was noted that a surrogate decision-maker may not be a single designated individual but could instead comprise multiple individuals or families. In addition, the description of advanced directives was revised to include both oral and written forms rather than limiting them to written documentation.

**Table 5. Validation of modifications by experts.**

| Program components | Revision items | I-CVI |
|---|---|---|
| Care Recipients Under the Intervention Program | The program content was restructured into a tiered forma | 0.83 |
| | Revised on "Initial screening items" and "Basic Information Related to the Quality of End-of-Life Care." | 0.83 |
| Provision of evidence-based symptom management | Methods (Timing, Participants, Procedures, etc.) | 1.00 |
| | Use of Japanese Evidence | 1.00 |
| | Revisions to Wording | 1.00 |
| Multidisciplinary Bedside Conferences | Methods (Timing, Participants, Procedures, etc.) | 1.00 |
| | Use of Japanese Evidence | 0.83 |
| Family Conference | Methods (Timing, Participants, Procedures, etc.) | 1.00 |
| | Revisions to Wording | 0.83 |
| End-of-Life Care | Methods (Timing, Participants, Procedures, etc.) | 0.83 |
| | Use of Japanese Evidence | 1.00 |

The finalized Nurse-Led Intervention Program to Improve Palliative Care and Quality of Dying and Death in the Intensive Care Unit is presented in S3 Text, and the screening and conference items are provided in S4 Text.

## Discussion

This study utilized existing evidence identified through SR/MA and ScR to develop a nurse-led intervention program to improve palliative care and QODD in the ICU. Given the lack of interventions targeting symptom management [3,7] despite its recognized importance in improving QODD in ICU patients, a provisional intervention program was initially developed with a focus on symptom management. This provisional program underwent internal and external validation through FGI with experts, resulting in a revised version that aligned with the feedback obtained. The FGI emphasized the importance of initiating symptom management as early as possible after ICU admission in Japan. Furthermore, the timing, methods, and participants involved in multidisciplinary bedside and family conferences were adjusted to enhance the feasibility of the program for the Japanese healthcare system. The program was structured in a ladder format to ensure timely and appropriate delivery of care to target patients. Additional components were incorporated, including the preparation of a therapeutic environment for patients near EOL and consideration of transfer to a general ward for bereavement care. The validity of the revised program was assessed via a web-based expert survey, to confirm if the content validity for all revised items was I-CVI ≥ 0.78. The intervention program was developed through a scientifically rigorous process in accordance with the 6SQuID framework. This approach enables the adaptation and development of evidence-based intervention programs in countries with different cultural contexts and healthcare systems.

Symptom management in the ICU remains suboptimal. An intervention program designed to facilitate consistent care delivery, independent of the providers' knowledge or attitudes, may enhance symptom management. Patients in the ICU frequently experience unrelieved distress symptoms [5], which are associated with lower QODD [3]. The conflation of "palliative care" and "EOL care" [14] may contribute to the underutilization of symptom palliation in ICUs. A previous intervention to improve ICU palliative care based on the self-efficacy theory did not significantly improve QODD or pain assessment [15], despite aiming to change intensivists' knowledge and behavior [16]. In contrast, the present program uses a structured booklet detailing the care content and implementation methods tailored to patients' conditions. This system-based approach promotes consistent practice regardless of individual self-efficacy or behavioral changes, thus potentially improving ICU symptom management and EOL care quality.

A system that facilitates effective communication between healthcare professionals and family members is essential to improve the quality of palliative and EOL care in the ICU. However, in the Japanese healthcare system, communication through bedside conferences and family meetings may not be adequately implemented without structured interventions. Closed ICU systems, which are common internationally [17], have demonstrated improved interprofessional communication [18] and outcomes [19,20]. In contrast, Japan has a higher prevalence of open ICUs (65.6% versus 17.1% internationally) [21], thus reflecting its ICU admission patterns [22]. Our findings highlight the challenges of conducting timely multidisciplinary meetings involving physicians. To address this issue, a ladder-based program was designed with clear criteria and care content for each stage to facilitate timely interventions and role-sharing, even in open ICU systems. The ladder approach presents staged interventions that are aligned with the patient's condition and decision-making phase. Thus, it enables the delivery of appropriate care at the right time, even with limited human resources [23]. Expert feedback also prompted a more flexible scheduling of meetings, which was expected to improve feasibility and staff participation.

The care environment during bereavement influences the quality of EOL care This program includes components to improve bereavement settings, such as easing visitation restrictions and enabling transfers to general wards. A previous study on preferred death [24] identified eight key elements, including "dying in a favorite place" and "attending the deathbed," thus underscoring the importance of setting and family presence. The ICU-QODD also highlighted family presence [25], thus affirming the need to optimize visitation policies. Although home-based EOL care for ICU patients has been reported [26,27], these reports are limited to Japan [28]. Owing to the challenges in discharging critically ill patients and the limitations in the

healthcare infrastructure, home care was not included in the standardized program. Given that preferences for home-based care are shaped by patients' wishes, cultural norms, and systemic constraints, further studies are warranted.

## Strengths and limitations

This intervention program was developed using scientifically rigorous methodology and was adapted to the Japanese healthcare system and cultural context. However, this study has some limitations. First, most of the literature on program development originates outside Japan, with limited domestic evidence. The program was modified using expert FGIs to align it with the Japanese context. Second, the patients and their families were not included as stakeholders in the development process. Nevertheless, given that the intervention was designed for the ICU setting, and revisions were based on input from nurses and physicians who play central roles, the program's feasibility and acceptability may have been enhanced [29]. Future research should assess these aspects through studies involving patients and their families. Finally, the program's effectiveness was not evaluated. Further implementation studies are required to examine the feasibility, acceptability, and impact of this program.

## Conclusion

We developed a nurse-led intervention program to improve palliative care and QODD in ICUs following the 6SQuID framework. The program centers on symptom management and includes three additional core components: multidisciplinary bedside conferences, family conferences, and structured end-of-life care. The content and external validity of the program were established through a literature review and input from an expert panel. Therefore, this program has potential utility in enhancing the quality of palliative and EOL care in Japanese ICUs. Furthermore, by adhering to a scientifically rigorous developmental process, this approach offers a model for adapting evidence-based interventions to diverse cultural and healthcare contexts beyond Japan.

## Supporting information

**S1 Text. Survey instrument used for content validity indexes evaluation.**
(DOCX)

**S2 Text. Item-level content validity indexes ratings.**
(DOCX)

**S3 Text. Nurse-led intervention program to improve the palliative care and quality of dying and death in intensive care unit.**
(DOCX)

**S4 Text. Screening and conference items.**
(DOCX)

## Acknowledgments

We thank the expert panel members for their support and cooperation in the development of the intervention program. We acknowledge Editage (www.editage.com) for assistance with English language editing.

## Author contributions

**Conceptualization:** Kazuaki Naya, Hideaki Sakuramoto.

**Data curation:** Kazuaki Naya.

**Formal analysis:** Kazuaki Naya.

**Funding acquisition:** Kazuaki Naya.

**Investigation:** Kazuaki Naya.

**Methodology:** Kazuaki Naya, Hideaki Sakuramoto.

**Project administration:** Kazuaki Naya.

**Resources:** Kazuaki Naya.

**Software:** Kazuaki Naya.

**Supervision:** Hideaki Sakuramoto.

**Validation:** Kazuaki Naya.

**Visualization:** Kazuaki Naya.

**Writing – original draft:** Kazuaki Naya.

**Writing – review & editing:** Kazuaki Naya, Hideaki Sakuramoto.

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
