## [Decision Letter · Decision Letter 0]

4 Dec 2025

Dear Dr. Sakuramoto,

We look forward to receiving your revised manuscript.

Kind regards,

JONATHAN BAYUO, PhD

Academic Editor

PLOS ONE

Journal Requirements:

This study was supported by a Grant-in-Aid for Scientific Research (C) from the Japan Society for the Promotion of Science (Grant number 25K13875).

N/A

5. Please remove all personal information, ensure that the data shared are in accordance with participant consent, and re-upload a fully anonymized data set.Note: spreadsheet columns with personal information must be removed and not hidden as all hidden columns will appear in the published file.Additional guidance on preparing raw data for publication can be found in our Data Policy (https://journals.plos.org/plosone/s/data-availability#loc-human-research-participant-data-and-other-sensitive-data) and in the following article: ) and in the following article: http://www.bmj.com/content/340/bmj.c181.long..

7. We note that Supporting files [S1 text.docx] includes an image of a [patient / participant / in the study].

Reviewers' comments:

Reviewer's Responses to Questions

**Comments to the Author**

1. Is the manuscript technically sound, and do the data support the conclusions?

Reviewer #1: Yes

Reviewer #2: Yes

2. Has the statistical analysis been performed appropriately and rigorously?

Reviewer #1: Yes

Reviewer #2: Yes

3. Have the authors made all data underlying the findings in their manuscript fully available?

Reviewer #1: Yes

Reviewer #2: Yes

4. Is the manuscript presented in an intelligible fashion and written in standard English?

Reviewer #1: Yes

Reviewer #2: Yes

Reviewer #1: Dear Author,

Congratulations for your great contribution to improvement of quality nursing care in palliative care.

General comments: The article presents an important contribution to the care of patients under palliative care in the ICU. The manuscript is well structured and described. However, there are some areas that require further description and clarification.

Abstract: please give a short background of the study and use paragraph to make it more organized.

Introduction: You have provided the background of your study and identified the gap in previous studies in the similar field

Methods: Did you yourself conducted both the scoping review, and systematic review and meta analysis?

Step 4, Can you explain the intervention program in detail?

Line 119: why did you exclude the participant who had experienced the loss of a family member within the next past 6 months?

Results: Great! you have clearly explained the details you undertook in each step.

Discussion: well written

Conclusion: Please restate the main aspects of the intervention program

Reviewer #2: The manuscript is technically sound. It clearly and coherently describes a systematic process for developing a complex intervention based on the 6SQuID framework.

The steps of problem identification, systematic review, exploratory review, program development, and expert validation are well documented.

The data presented (citations, tables, qualitative descriptions) adequately support the conclusions regarding the content validity and cultural relevance of the program.

The methodology is coherent and consistent with the objectives.

The design is framed within content development and validation; the use of the Content Validity Index (CVI) is appropriate, correctly calculated, and reported.

The methodology follows classic recommendations (Lynn, 1986), using a threshold of ≥0.78 to accept items validated by experts.

No additional inferential statistics are required for this type of study; the analysis is sufficient and rigorous for the stated objectives. The authors declare that all relevant data are included in the manuscript and supplementary files, which adequately complies with the PLOS data policy.

The information provided is consistent with the expected elements for qualitative and methodological development studies.

The manuscript is well-written, with clear and structured language.

There are some minor writing and flow details that could be corrected (e.g., redundancies, long sentences), but none affect overall comprehension.

The quality of the English is adequate for publication, although a final light revision is recommended.

Strengths of the manuscript:

1. The study uses a recognized framework (6SQuID) that provides methodological rigor.

2. The integration of prior evidence (SR/MA and Scoping Review) and expert validation is well achieved.

3. The culturally adapted approach for the Japanese context is a significant contribution to the literature on palliative care in the ICU.

4. The ladder structure of the program is logical, clear, and potentially applicable in different clinical settings.

Areas for Improvement

1. Clarity in the transition between ladder levels: Although adjusted based on expert feedback, it would be helpful to explain in more detail the operational criteria for moving between levels, especially between Ladder II and Ladder III.

2. Stronger justification for the inclusion of non-terminal patients: While explained in the discussion, reinforcing this point in the introduction would help avoid misinterpretations about the scope of the program.

3. Describe how qualitative saturation was managed in the FGI phase, given that six experts participated. This is not essential but would increase methodological transparency.

4. English fluency: Some paragraphs could benefit from grammatical simplification, especially in the Methods section. This does not affect comprehension but could increase clarity for international audiences.

Overall Review Conclusion

The manuscript is robust, rigorous, and provides a relevant intervention program for critical care settings, with the potential to be replicated in other cultural contexts. I recommend publishing it after addressing the minor formatting issues.

.

Reviewer #1: **Yes:** Mira Adhikari BaralMira Adhikari BaralMira Adhikari BaralMira Adhikari Baral

Reviewer #2: No

---

## [Author Response · Author response to Decision Letter 1]

19 Jan 2026

Dear Dr. JONATHAN BAYUO

Thank you the opportunity to revise and resubmit our manuscript, “Development and external validity of a nurse-led intervention program to improve palliative care and quality of dying and death in intensive care unit,” to PLOS One. We sincerely appreciate the editor’s handling of our submission and the reviewers’ careful and constructive evaluations, which have been invaluable in strengthening the manuscript.

In response to the reviewers’ comments, we have revised the manuscript accordingly and provide a detailed, point-by-point response below outlining how each concern was addressed. We believe that these revisions have substantially improved the clarity, rigor, and transparency of the work, and we respectfully resubmit the manuscript for your further consideration.

Reviewer #1:

Comment 1. Abstract: please give a short background of the study and use paragraph to make it more organized.

Response:

To address this, we have revised the Abstract to include a brief background of the study and reorganized it into clearly defined paragraphs corresponding to the Introduction, Methods, Results, and Conclusion sections to improve clarity and readability.

Comment 2. Introduction: You have provided the background of your study and identified the gap in previous studies in the similar field

Response:

We appreciate the reviewer’s recognition that the introduction adequately provides the background of the study and clearly identifies the research gaps in previous studies.

Comment 3. Methods: Did you yourself conducted both the scoping review, and systematic review and meta analysis?

Response:

Yes, we conducted the scoping review, systematic review, and meta-analysis; however, this was not clearly stated in the original manuscript. We have revised the Methods section to clarify this point and added relevant references.

Page 4, line 87-91: Initially, we conducted a systematic review and meta-analysis (SR/MA) [3] and a scoping review (ScR) [7]. The findings from these reviews were synthesized and applied in Steps 1–3. Step 4 involved drafting a preliminary version of the intervention program and assessing its internal and external validity. Based on the results, the program was revised and finalized.

Comment 4. Step 4, Can you explain the intervention program in detail?

Response:

To address the comment, in the revised manuscript, we have provided a detailed description of the intervention program, including both the original version and revisions made based on expert feedback. Initially, the program was designed as a time- and condition-based flow, reflecting the clinical trajectories of critically ill patients in the ICU. Following expert feedback, we reorganized the intervention into a stepwise ladder-like format to clarify the operational criteria for transitions between levels and to provide a more structured guide for clinical practice. We have added a description of the original flow-based program, the rationale for the reorganization, and details of each ladder level (I–III) to the Methods section. This allows readers to understand both the original program and the modifications made to enhance clarity, feasibility, and applicability in the ICU setting.

Page 11, line 205-222: Based on these eight intervention strategies, we developed a preliminary nurse-led intervention program to improve palliative care and QODD in ICU. Rather than following a ladder-based approach, the intervention was designed with a time- and condition-dependent flow of care. Patients were screened based on their clinical status over time and subsequent steps in the intervention proceeded according to this screening, creating a structured flow throughout the ICU stay. A flow diagram was developed to visually guide the practice.

Given the complexity of transitioning from curative to EOL care in modern ICUs, it is necessary to identify appropriate patients at an early stage, rather than focusing only on those who are near death. Therefore, primary screening criteria were established to identify eligible patients from among all ICU admissions. Once identified, evidence-based symptom management was initiated on ICU admission. For these patients, multidisciplinary bedside conferences were held within 24 h to clarify the goals of care and discuss treatment strategies. This step enabled the early planning of family conferences and timely initiation of EOL care. Nurses closely involved in daily patient care serve as key agents of the program. Their roles in symptom assessment, patient identification, and coordination of multidisciplinary meetings were explicitly defined and formally integrated into the intervention procedures. The intervention program was compiled into a booklet including an intervention flow diagram and designed for distribution to the departments participating in the intervention.

Page 20-21, line 272-287: The primary screening items were revised in response to these comments, and the intervention process, which originally followed a time- and condition-based flow, was reorganized into a stepwise ladder-like format (Fig. 2). This allowed for the initiation of symptom management from the time of ICU admission without altering the overall target population specified in the intervention program.

Following this reorganization, explicit operational criteria were defined for transitions between ladder levels (Fig. 2). Ladder I was initiated for patients who required, or were highly likely to require, mechanical ventilation for more than 48 hours, or who were admitted emergently from a general ward. The transition from Ladder I to Ladder II was considered when the patient’s condition deteriorated or the response to treatment was judged to be poor. Nurses conducted this assessment during daily symptom management conferences. The transition from Ladder II to Ladder III was determined when life-saving treatment was judged difficult and the likelihood of death was extremely high. This decision was made by the nurses during multidisciplinary bedside conferences. In cases of rapid clinical deterioration or an extremely poor response to treatment, a direct transition from Ladder I to Ladder III was permitted. All screening and transition decisions were nurse-led and reassessed based on changes in the patient’s clinical condition.

Comment 5. Line 119: why did you exclude the participant who had experienced the loss of a family member within the past 6 months?

Response:

Individuals within six months of bereavement were excluded because early grief reactions and psychological distress are generally the most pronounced during this period. Because this intervention program focuses on improving the quality of dying and death, participation can evoke memories of recent losses and potentially exacerbate psychological distress. Therefore, to minimize the emotional burden and ethical risk to the participants, individuals who had experienced the loss of a family member within the past six months were excluded.

Page 6, line 133-137: (g) participants who had experienced the loss of a family member within the past six months. Since early bereavement reactions and psychological symptoms tend to be most pronounced within the first six months following loss [11], these individuals were excluded to minimize psychological distress and avoid triggering recollection of recent bereavement.

Comment 6. Results: Great! you have clearly explained the details you undertook in each step. Discussion: well written

Response:

Thank you very much for your positive and encouraging comments. They were very encouraging and motivating for our research team.

Comment 7. Conclusion: Please restate the main aspects of the intervention program

Response:

To address this, we have revised the conclusion to clearly restate the main aspects of the intervention program. Specifically, we have added a description that the program consists of symptom management as its core component, along with three additional key elements.

Page 29, line 445-452: We developed a nurse-led intervention program to improve palliative care and QODD in ICUs following the 6SQuID framework. The program centers on symptom management and includes three additional core components: multidisciplinary bedside conferences, family conferences, and structured end-of-life care. The content and external validity of the program were established through a literature review and input from an expert panel. Therefore, this program has potential utility in enhancing the quality of palliative and EOL care in Japanese ICUs. Furthermore, by adhering to a scientifically rigorous developmental process, this approach offers a model for adapting evidence-based interventions to diverse cultural and healthcare contexts beyond Japan.

Reviewer #2:

Comment 1. Clarity in the transition between ladder levels: Although adjusted based on expert feedback, it would be helpful to explain in more detail the operational criteria for moving between levels, especially between Ladder II and Ladder III.

Response:

We have added detailed explanations regarding the operational criteria for transitions between ladder levels in the Results section. Specifically, we describe the clinical conditions and decision criteria for transitions from Ladder I to II, from Ladder II to III, and from Ladder I directly to III. In addition, we clarified who performed the assessment and at what times. These explanations are consistent with the information presented in Figure 2.

Page 20-21, line 277-287: Following this reorganization, explicit operational criteria were defined for transitions between ladder levels (Fig. 2). Ladder I was initiated for patients who required, or were highly likely to require, mechanical ventilation for more than 48 hours, or who were admitted emergently from a general ward. The transition from Ladder I to Ladder II was considered when the patient’s condition deteriorated or the response to treatment was judged to be poor. Nurses conducted this assessment during daily symptom management conferences. The transition from Ladder II to Ladder III was determined when life-saving treatment was judged difficult and the likelihood of death was extremely high. This decision was made by the nurses during multidisciplinary bedside conferences. In cases of rapid clinical deterioration or an extremely poor response to treatment, a direct transition from Ladder I to Ladder III was permitted. All screening and transition decisions were nurse-led and reassessed based on changes in the patient’s clinical condition.

Comment 2. Stronger justification for the inclusion of non-terminal patients: While explained in the discussion, reinforcing this point in the introduction would help avoid misinterpretations about the scope of the program.

Response:

To address this, we have reinforced the rationale for including non-terminal patients in the intervention program by adding an explanation to the background section to clarify the scope of the program.

Page 3, line 63-72: In the ICU setting, patient prognosis is often uncertain at admission and may change over time. Many patients who die in the ICU initially receive aggressive, life-sustaining treatments with the expectation of recovery. Therefore, discussions regarding patient values, preferences, and advance directives should be initiated early after ICU admission and revisited as clinical conditions evolve. Inadequate symptom control is a major contributor to poor QODD [4], and ICU patients frequently experience significant unrelieved physical and psychological distress [5]. Accordingly, palliative care should be provided to all ICU patients regardless of their prognosis. Although ICU palliative care initially focused on EOL care, it is now understood to encompass symptom management, shared decision-making, and support for patients and their families throughout the course of the critical illness [6].

Comment 3. Describe how qualitative saturation was managed in the FGI phase, given that six experts participated. This is not essential but would increase methodological transparency.

Response:

Although qualitative saturation was not a predefined endpoint for the FGI, it was considered achieved when no new themes or suggestions relevant to refining the intervention program emerged during the later stages of the discussion. The six experts had diverse professional backgrounds relevant to ICU palliative and end-of-life care and their feedback converged on similar key issues, suggesting sufficient information redundancy. In addition, we verified the analysis results with the participants to ensure that their statements were accurately reflected and that no additional opinions were omitted. We have clarified this in the manuscript:

Page 6, line 138-147: Interviews were conducted remotely via video conferencing. Audio recordings were transcribed verbatim and reviewed multiple times. Common themes were identified and synthesized. Revisions of the preliminary program were based on expert feedback. Qualitative saturation was achieved when no new themes or suggestions relevant to the refinement of the intervention program emerged during the later stages of the FGI. To ensure trustworthiness of the content analysis, the data were reviewed repeatedly by two researchers and supervised by a qualitative research expert. The raw data were re-examined to ensure accuracy. Additionally, the analyzed data were re-examined and verified with the participants to ensure that their statements were accurately reflected and that no additional opinions were omitted. The revised version was defined as the “modified intervention program.” The interview guidelines are presented in Table 1.

Comment 4. English fluency: Some paragraphs could benefit from grammatical simplification, especially in the Methods section. This does not affect comprehension but could increase clarity for international audiences.

Response:

Thank you for your comments. We have revised the manuscript for clarity and readability, particularly in the Methods section. The manuscript has been professionally edited by Editage to ensure grammatical accuracy and improve clarity for an international audience.

Comment 5. Overall Review Conclusion: The manuscript is robust, rigorous, and provides a relevant intervention program for critical care settings, with the potential to be replicated in other cultural contexts. I recommend publishing it after addressing the minor formatting issues.

Thank you for your positive evaluation and the constructive feedback. We are encouraged to find the manuscript robust and the intervention program relevant and potentially generalizable. We have addressed the minor formatting issues, as suggested. We look forward to hearing from you and would be happy to make further changes,

---

## [Decision Letter · Decision Letter 1]

6 Feb 2026

Dear Dr. Sakuramoto,

Thank you for submitting your manuscript to PLOS ONE. After careful consideration, we feel that it has merit but does not fully meet PLOS ONE’s publication criteria as it currently stands. Therefore, we invite you to submit a revised version of the manuscript that addresses the points raised during the review process.

We look forward to receiving your revised manuscript.

Kind regards,

JONATHAN BAYUO, PhD

Academic Editor

PLOS One

Journal Requirements:

Reviewers' comments:

Reviewer's Responses to Questions

**Comments to the Author**

Reviewer #2: All comments have been addressed

2. Is the manuscript technically sound, and do the data support the conclusions?

Reviewer #2: Yes

3. Has the statistical analysis been performed appropriately and rigorously?

Reviewer #2: Yes

4. Have the authors made all data underlying the findings in their manuscript fully available?

Reviewer #2: No

5. Is the manuscript presented in an intelligible fashion and written in standard English?

Reviewer #2: Yes

Reviewer #2: 1) Is the manuscript technically sound, and do the data support the conclusions?

Answer: Yes, it is generally technically sound; however, some conclusions require nuance to align with the actual scope of the data.

• Technical soundness: The development and expert validation are described with clear procedures (FGI + web survey) and criteria for qualitative rigor (review by two researchers, expert supervision, verification with participants, qualitative saturation).

• Support for conclusions: The data support the development and refinement of a program with content validity according to experts (CVI), but the abstract statement can be interpreted as a claim of already demonstrated effectiveness/consistency (“enables a more consistent delivery… regardless of individual providers’ knowledge or attitudes”). This exceeds what can be concluded without an implementation/effectiveness evaluation.

Specific suggestion: Change to language such as “designed for / could facilitate / has the potential to” and maintain the conclusion focused on expert validity.

2) Was the statistical analysis performed adequately and rigorously?

Answer: Yes, for the study's objective (expert validation), the quantitative analysis is appropriate; however, it is limited to content validity (it does not assess effectiveness).

• The "statistical" component is primarily the Content Validity Index (CVI) using Lynn's method, a 4-point scale, calculation of the I-CVI, a threshold ≥ 0.78, and iterative modification until the threshold is reached for all items. This is explicitly described and is consistent with standard practices for content validity.

• No complex inferential analyses were observed (because the design does not require them). In this sense, the quantitative rigor is appropriate for this type of study.

3) Have the authors made all the underlying data for the findings in their manuscript publicly available?

Answer: Partially / not fully verifiable with the information provided in the review PDF.

• • In the editorial form, they state: “All relevant data are within the manuscript and its Supporting Information files.”

But for a study with FGI + CVI, “underlying data” would typically include at least:

a matrix/sheet of CVI ratings per item (or complete table), an instrument (survey, interview guide), and, if transcripts cannot be shared, some level of auditable qualitative evidence (e.g., a table of topics with anonymized extracts or a codebook).

In the material I reviewed, I can confirm that the statement is there, but I cannot confirm that these inputs are actually accessible as complete underlying data based solely on the system phrase.

Suggestion: The authors should explicitly specify which data are in the Supporting Information (e.g., “CVI item-level ratings as S1 Dataset,” “survey instrument as S1 Appendix,” etc.).

4) Is the manuscript presented in an intelligible manner and written in standard English? Answer: Yes, it is intelligible and in standard English; minor style/consistency adjustments will be necessary.

• The authors declare that they have reviewed clarity and readability, especially in the Methods section, and that the manuscript was professionally edited by Editage for grammatical accuracy and international clarity.

• As a minor detail, traces of editing are still visible (e.g., small typos/duplications in some lines of the Methods section), but these do not impede comprehension.

.

Reviewer #2: No

---

## [Author Response · Author response to Decision Letter 2]

4 Mar 2026

Dear Dr. JONATHAN BAYUO

Thank you the opportunity to revise and resubmit our manuscript, “Development and external validity of a nurse-led intervention program to improve palliative care and quality of dying and death in intensive care unit,” to PLOS One. We sincerely appreciate the editor’s handling of our submission and the reviewers’ careful and constructive evaluations, which have been invaluable in strengthening the manuscript.

In response to the reviewers’ comments, we have revised the manuscript accordingly and provide a detailed, point-by-point response below outlining how each concern was addressed. We believe that these revisions have substantially improved the clarity, rigor, and transparency of the work, and we respectfully resubmit the manuscript for your further consideration.

Reviewer #2:

Comment 1. Support for conclusions: The data support the development and refinement of a program with content validity according to experts (CVI), but the abstract statement can be interpreted as a claim of already demonstrated effectiveness/consistency (“enables a more consistent delivery… regardless of individual providers’ knowledge or attitudes”). This exceeds what can be concluded without an implementation/effectiveness evaluation.

Response:

To address this, we have revised the wording in the Abstract to avoid implying demonstrated effectiveness and to better align the conclusions with the scope of this development and content validation study. Specifically, following the reviewer’s suggestion, we replaced “enables a more consistent delivery of end-of-life care in the ICU” with “was designed to facilitate more consistent delivery of end-of-life care in ICU.”

For consistency, we also revised similar wording in the Discussion section.

Page 27, line 399-401: Symptom management in the ICU remains suboptimal. An intervention program designed to facilitate consistent care delivery, independent of the providers’ knowledge or attitudes, may enhance symptom management.

Comment 2. In the editorial form, they state: “All relevant data are within the manuscript and its Supporting Information files.”

But for a study with FGI + CVI, “underlying data” would typically include at least: a matrix/sheet of CVI ratings per item (or complete table), an instrument (survey, interview guide), and, if transcripts cannot be shared, some level of auditable qualitative evidence (e.g., a table of topics with anonymized extracts or a codebook).

In the material I reviewed, I can confirm that the statement is there, but I cannot confirm that these inputs are actually accessible as complete underlying data based solely on the system phrase.

Response:

Thank you for your important comment regarding data availability and transparency.

In response, we have ensured that all relevant underlying data are accessible through the Supporting Information files. Specifically, we have added:

S1 Text: Survey instrument used for content validity index (CVI) evaluation

S2 Text: Item-level CVI ratings for each expert

Regarding the FGI data, full interview transcripts cannot be publicly shared due to ethical restrictions and the potential identifiability of participants. However, we have provided a structured summary of themes and representative anonymized excerpts within the manuscript to ensure auditability while protecting confidentiality.

We believe these additions ensure that all relevant underlying data supporting the study findings are now fully accessible in accordance with the journal’s data availability policy.

Comment 3. As a minor detail, traces of editing are still visible (e.g., small typos/duplications in some lines of the Methods section), but these do not impede comprehension.

Response:

Thank you for this comment. We carefully reviewed the Methods section and corrected minor typographical errors, punctuation issues, and wording duplications throughout the manuscript.

---

## [Decision Letter · Decision Letter 2]

22 Mar 2026

Development and external validity of a nurse-led intervention program to improve palliative care and quality of dying and death in intensive care unit

PONE-D-25-47956R2

Dear Dr. Sakuramoto,

We’re pleased to inform you that your manuscript has been judged scientifically suitable for publication and will be formally accepted for publication once it meets all outstanding technical requirements.

Kind regards,

JONATHAN BAYUO, PhD

Academic Editor

PLOS One

Additional Editor Comments (optional):

Thanks to the authors for thoughtfully addressing the comments raised.

Reviewers' comments:

Reviewer's Responses to Questions

**Comments to the Author**

Reviewer #2: All comments have been addressed

2. Is the manuscript technically sound, and do the data support the conclusions?

Reviewer #2: Yes

3. Has the statistical analysis been performed appropriately and rigorously?

Reviewer #2: Yes

4. Have the authors made all data underlying the findings in their manuscript fully available?

Reviewer #2: Yes

5. Is the manuscript presented in an intelligible fashion and written in standard English?

Reviewer #2: Yes

Reviewer #2: (No Response)

.

Reviewer #2: **Yes:** Claudia Barría-SandovalClaudia Barría-SandovalClaudia Barría-SandovalClaudia Barría-Sandoval

---

## [Editor Report · Acceptance letter]

PONE-D-25-47956R2

PLOS One

Dear Dr. Sakuramoto,

I'm pleased to inform you that your manuscript has been deemed suitable for publication in PLOS One. Congratulations! Your manuscript is now being handed over to our production team.

Kind regards,

on behalf of

Dr. JONATHAN BAYUO

Academic Editor

PLOS One